# The role of IMP dehydrogenase 2 in Inauhzin-induced ribosomal stress

**Qi Zhang[1,2], Xiang Zhou[1,2], RuiZhi Wu[1,2], Amber Mosley[3], Shelya X Zeng[1,2], Zhen Xing[4], Hua Lu[1,2]***

[1]Department of Biochemistry and Molecular Biology, Tulane University School of Medicine, New Orleans, United States; [2]Tulane Cancer Center, Tulane University School of Medicine, New Orleans, United States; [3]Department of Biochemistry and Molecular Biology, Indiana University School of Medicine, Indianapolis, United States; [4]Department of Molecular and Cellular Oncology, University of Texas MD Anderson Cancer Center, Houston, United States

**Abstract** The 'ribosomal stress (RS)-p53 pathway' is triggered by any stressor or genetic alteration that disrupts ribosomal biogenesis, and mediated by several ribosomal proteins (RPs), such as RPL11 and RPL5, which inhibit MDM2 and activate p53. Inosine monophosphate (IMP) dehydrogenase 2 (IMPDH2) is a rate-limiting enzyme in de novo guanine nucleotide biosynthesis and crucial for maintaining cellular guanine deoxy- and ribonucleotide pools needed for DNA and RNA synthesis. It is highly expressed in many malignancies. We previously showed that inhibition of IMPDH2 leads to p53 activation by causing RS. Surprisingly, our current study reveals that Inauzhin (INZ), a novel non-genotoxic p53 activator by inhibiting SIRT1, can also inhibit cellular IMPDH2 activity, and reduce the levels of cellular GTP and GTP-binding nucleostemin that is essential for rRNA processing. Consequently, INZ induces RS and the RPL11/RPL5-MDM2 interaction, activating p53. These results support the new notion that INZ suppresses cancer cell growth by dually targeting SIRT1 and IMPDH2.

*For correspondence: hlu2@tulane.edu

**Competing interests:** The authors declare that no competing interests exist.

**Reviewing editor**: Carol Prives, Columbia University, United States

## Introduction

With ~22 million people living with cancers that are highly associated with alterations of multiple molecules and pathways, it is important to develop a multiple molecules-targeted therapy that can effectively kill cancer cells. The tumor suppressor p53 pathway is one such a target because nearly all cancers show defects in this pathway. Approximately 50% of human cancers have mutations in the TP53 gene itself, while the rest of them harbor functionally inactive p53 proteins, because active p53 can trigger cell growth arrest, apoptosis, autophagy, and/or senescence, which are detrimental to cancer cells (*Vogelstein et al., 2000*; *Vousden and Prives, 2009*), and impede cell migration, metabolism, and/or angiogenesis. A major mechanism for functional inactivation of p53 is through overexpression of two chief p53 suppressors, MDM2 and MDMX, which work together to inactivate p53 by directly interacting with p53, inhibiting its transcriptional activity and mediating its ubiquitin dependent degradation (*Wade et al., 2010*; *Huang et al., 2011*; *Tollini and Zhang, 2012*). This MDM2/MDMX-mediated p53 degradation is also facilitated by SIRT1, a nicotinamide adenine dinucleotide (NAD+)-dependent deacetylase (*Vaziri et al., 2001*; *Cheng et al., 2003*). SIRT1 is highly expressed in human cancers due to down regulation of another p53 target tumor suppressor called hypermethylated in cancer-1 (HIC-1) (*Chen et al., 2005*).

Our previous study identified a small molecule named Inauhzin (INZ) that effectively inhibits SIRT1 activity and induces p53 acetylation, leading to the increase of p53 level and activity (*Zhang et al., 2012b*). Consequently, INZ induces p53-dependent apoptosis and senescence in various p53-wild

**eLife digest** Cancer develops when cells lose the ability to control their own growth. About half of cancerous tumors carry a dysfunctional version of a protein called p53, while the other half have defects in proteins that are important for p53's production and function. When a healthy cell is exposed to damaging chemicals or agents, the p53 protein triggers responses that are aimed at repairing the damage. However, if these attempts fail, p53 causes the damaged cell to essentially destroy itself.

As defects in p53-controlled processes cause cells to grow unrestrictedly and can lead to cancer, it is a very attractive target for cancer therapies. Cancer drug developments have focused on both targeting p53 directly and targeting the proteins that work with p53. Two proteins called Mdm2 and SIRT1 are of particular interest. Mdm2 binds to, inactivates, and leads to the degradation of p53. SIRT1 can modify p53 and make it more accessible to Mdm2, and is often found in very high levels in cancer cells.

In 2012, researchers identified Inauhzin as a small molecule that could potentially be used to treat tumors that still have a functional version of the p53 protein. Inauhzin was thought to work by inhibiting SIRT1, which increases p53 levels—probably through its effects on Mdm2. This restores the cell's ability to control its growth and to die if it is irreparably damaged. However, not all of this small molecule's effects on cells can be explained by its interaction with SIRT1.

Now Zhang et al., including some of the researchers involved in the 2012 work, have investigated whether Inauhzin also interacts with other proteins in the cell; and Inauhzin was revealed to bind an enzyme called IMPDH2. This enzyme is involved in making GTP—a small molecule that is involved in many important processes in living cells. Zhang et al. demonstrated that Inauhzin's effect on the IMPDH enzyme triggered a response that did not involve the SIRT1 protein, and that ultimately led to a decrease in Mdm2 activity and restored p53 activity.

Cancer treatments often include a combination of drugs that target different proteins with the goal of reducing the likelihood of a tumor becoming resistant to the treatment. Inauhzin's effect on two different proteins that lead to p53 activation not only increases its potency, but also makes it less likely that drug resistance will develop.

type human cancer cells, such as H460, and HCT116 by inducing the expression of p53-dependent transcriptome (*Liao et al., 2012*). INZ markedly inhibits the growth of H460 or HCT116 xenograft tumors, but is not toxic to normal cells and tissues. Also, INZ sensitizes the anti-cancer effect of cisplatin, doxorubicin, or Nutlin-3 (an MDM2 inhibitor) as tested in xenograft cancer models (*Zhang et al., 2012c*; *Zhang et al, 2013*). Thus, this small molecule presents as a promising contender for a molecule-targeted anti-cancer therapy. Since its discovery, we have optimized INZ (*Zhang et al., 2012a*) and determined additional cellular proteins that INZ might target via a set of biochemical, proteomic, and cell-based analyses. As detailed below, our study unveils inosine monophosphate (IMP) dehydrogenase 2 (IMPDH2) as a novel cellular target of INZ.

## Results and Discussion

### Identification of IMPDH2 as a cellular target of INZ

IMPDH is the key metabolic enzyme supplying guanine nucleotides to a cell as the first and rate-limiting enzyme of de novo GTP biosynthesis by catalyzing NAD+-dependent oxidation of IMP to xanthosine monophosphate (XMP) (*Zimmermann et al., 1995*; *Zhang et al., 1999*). IMPDH2 is the predominant isoform among its two isoenzymes, and often highly expressed in proliferating cells and neoplastic tissues (*Ishitsuka et al., 2005*; *Gu et al., 2003*), correlated to drug resistance, and thus has been used as a validated target for immunosuppressive (mycophenolic acid [MPA] [*Sintchak et al., 1996*] and mizoribine [*Gan et al., 2003*]), antiviral (ribavirin [*Prosise et al., 2002*]), and cancer-chemotherapeutic development [tiazofurin] (*Malek et al., 2004*; *Gu et al., 2005*; *Chen and Pankiewicz, 2007*; *Borden and Culjkovic-Kraljacic, 2010*).

Interestingly, by performing a biotin-INZ avidin affinity purification coupled with mass spectrometry (MS) analysis, we identified IMPDH2 as one of the top candidate proteins that INZ specifically

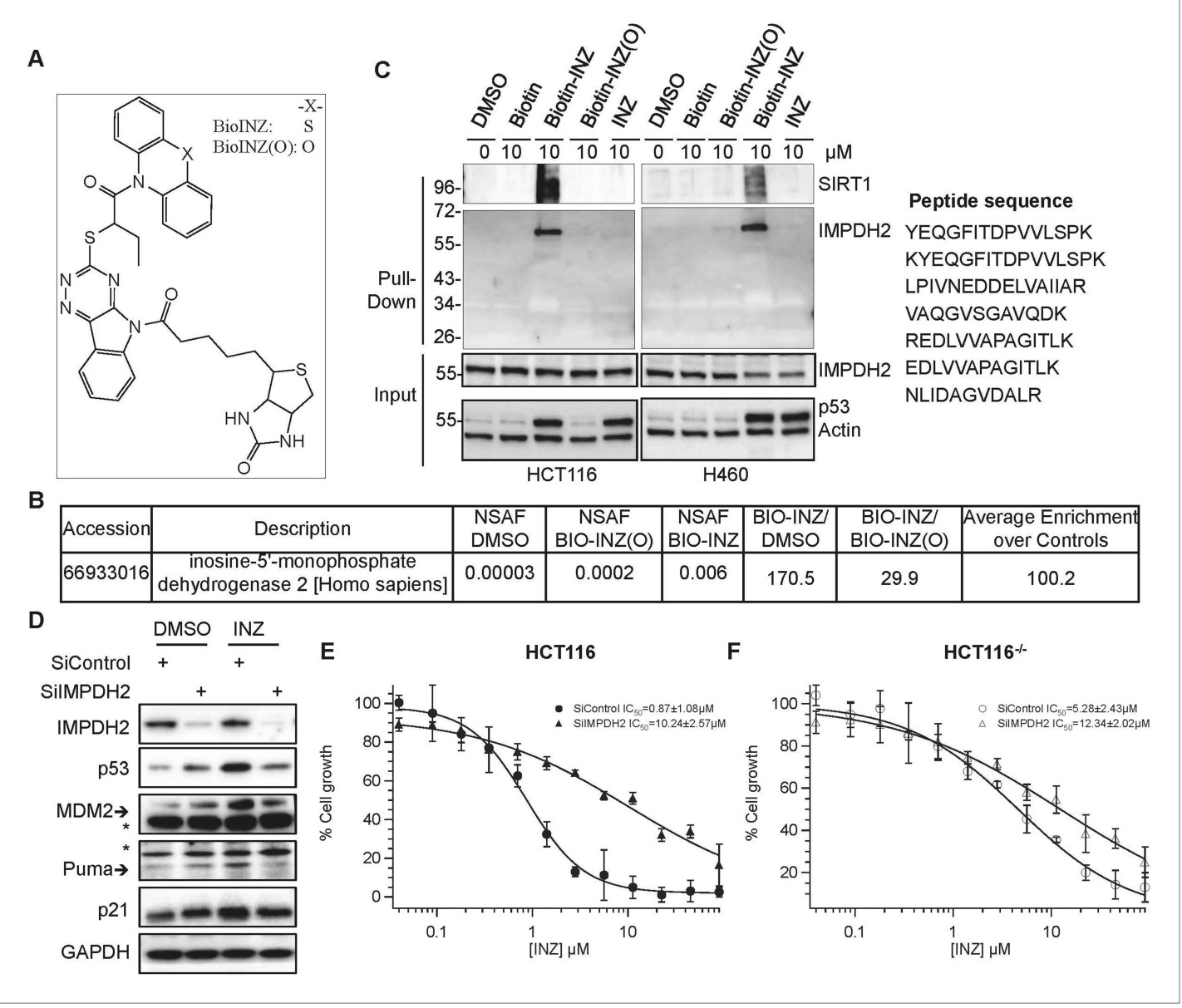

**Figure 1**. Identification of IMPDH2 as a potential target of INZ. (**A**) Structure of INZ analogs conjugated with Biotin (Biotin-INZ) used for INZ target identification, the oxygen-substituent (Biotin-INZ (O)) as a negative control. (**B–C**) Cells were treated with indicated compounds individually for 18 hr. Cleared cell lysates were incubated with NeutrAvidin beads and washed. The samples from HCT116 cells were then in-beads digested for MS analysis. NSAF: normalized spectral abundance factor. Samples were also resolved by SDS-PAGE and subjected to IB with indicated antibodies. (**D–F**) Knockdown of IMPDH2 alleviates INZ induction of p53. HCT116 cells were transfected with scrambled siRNA or IMPDH2 siRNA. 18 hr prior to harvesting, cells were treated with 2 μM INZ and harvested for IB with indicated antibodies (**D**). HCT116 and HCT116$^{-/-}$ cells were exposed to INZ for 72 hr and evaluated by WST cell growth assays (**E** and **F**). The $IC_{50}$ values of INZ in the scrambled siRNA and IMPDH2 siRNA transfected cells are 0.87 ± 1.08 μM and 10.24 ± 2.57 μM for HCT116 cells, and 5.28 ± 2.43 μM and 12.34 ± 2.02 μM for HCT116$^{-/-}$ cells, respectively (Mean ± SD, n = 3).

targets in cancer cells. Biotinylated INZ analogs (*Figure 1A*) were synthesized for these analyses. Here, Biotin-INZ was as active as INZ (*Zhang et al., 2012b*; *Zhang et al., 2012a*), while Biotin-INZ (O) was inactive and thus used as a negative control. Comparison of the most abundant proteins based on normalized spectral abundance factor (NSAF) in the cells treated Biotin-INZ vs DMSO or Biotin-INZ (O) revealed the high enrichment of IMPDH2 proteins in the former (*Figure 1B*) with enriched IMPDH2 peptides shown in *Figure 1C*. This result was firmly validated by immunoblot (IB) analysis of the pulled down proteins, as IMPDH2 was specifically brought down with Biotin-INZ, as well as together with our

previously identified SIRT1, but not Biotin-INZ (O) or other controls, in both H460 and HCT116 cells (*Figure 1C*).

To test if IMPDH2 is required for INZ activation of p53, we knocked down IMPDH2 in HCT116 cells with specific siRNAs in the presence or absence of INZ. As shown in *Figure 1D*, knockdown of IMPDH2 impeded INZ-induced p53 activation as indicated by the reduction of INZ-induced p53, p21, MDM2, and Puma levels. Consistently, the growth inhibition by INZ was compromised as the $IC_{50}$ value for INZ in cell growth analysis decreased by almost ~10-fold when IMPDH2 was knocked down (*Figure 1E*). Knockdown of IMPDH2 in HCT116 cells also conveyed much more significant effect on compromising the cytotox-icity of INZ compared to p53 null HCT116 (HCT116$^{-/-}$) cells (*Figure 1F*), indicating that INZ suppresses cancer cell growth mainly by targeting IMPDH2 in the cells and consequently activating the p53 pathway.

Although Biotin-INZ was associated with cellular IMPDH2 (*Figure 1*), INZ did not appear to affect the activity of the purified enzyme (date not shown). This discrepancy could be due to differences between the recombinant IMPDH2 in vitro and its native form in cells, as the latter could be regulated via post-translational modifications or partner proteins in cells, or INZ might mimic a nucleoside and be phosphorylated by a kinase in cells to target IMPDH2. These results also suggest that INZ might not directly bind to the active site of this enzyme. These possibilities remain to be addressed in the future.

## INZ causes the depletion of nucleostemin and consequent ribosomal stress

Our previous study showed that inhibition of IMPDH2 activity by MPA leads to RS and consequent p53 activation by reducing the level of nucleostemin (NS) (*Dai et al., 2008*; *Lo et al., 2012*), a nucleolar GTP-binding protein important for rRNA processing (*Tsai and McKay, 2005*; *Lo et al., 2012*). The association of INZ with IMPDH2 suggested that INZ might have a similar effect. As shown in *Figure 2A*, INZ, but not INZ(O), indeed significantly reduced NS protein levels, which was inversely correlated with the INZ induction of p53, p21, MDM2 and cleaved PARP. This result was further confirmed by immunofluorescence staining, as INZ, but not INZ(O), led to apparent decrease of nucleolar NS (*Figure 2B*). This decrease was due to the reduction of NS's half-life from >10 hr to <6 hr, as shown in *Figure 2C*, but NS mRNA level did not alter (data not shown). This result, also repeated in HCT116 cell lines (data not shown), demonstrates that INZ can destabilize cellular NS.

Next, we tested if the depletion of NS by INZ could induce the interaction of the RPL11 and RPL5 with MDM2, because we previously showed that the reduction of NS by MPA could induce RS and activate p53 by enhancing this interaction (*Sun et al., 2008*). As shown in *Figure 3A*, INZ, but not INZ(O), indeed enhanced the interaction of MDM2 with RPL5 and RPL11 in H460 cells by immunopre-cipitation (IP) using anti-MDM2 antibodies followed by IB (*Figure 3A*). The increased binding of MDM2 to L11 was true in a reciprocal co-IP using anti-L11 antibodies (*Figure 3A*). This result indicates that INZ-induced p53 activation involves suppression of MDM2 activity by the RPs, further supporting the RS-p53 response of INZ-treated cells.

To determine if RPL5 and RPL11 are required for INZ activation of p53, we performed a knockdown experiment. As expected, reduction of either RPL5 or RPL11 (data not shown) or both levels by siRNA markedly inhibited INZ-induced p53 level, compared to that in scrambled siRNA-transfected cells (*Figure 3B*). Consistently, knocking down RPL5 and RPL11 abrogated INZ-induced p21 and MDM2 levels (*Figure 3B*) and apoptosis (*Figure 3C*), indicating that RPL5 and RPL11 are required for INZ-induced p53 activation and apoptosis. Also, ribosome profile analysis followed by IB revealed that INZ significantly increases the levels of ribosome-free RPL5 and RPL11 (fractions 1–10), whereas it markedly reduces the level of polysomes (fractions 37–51) (*Figure 3D*), suggesting that INZ could suppress ribosome biogenesis and possibly protein translation. All together, these results demonstrate that INZ can acti-vate p53-dependent apoptosis by interfering with ribosome biogenesis through depletion of NS, causing RS, which then induces the release of ribosome-free RPL11 and RPL5 that bind to MDM2 and consequently inhibit its activity toward p53.

## INZ-mediated GTP depletion by targeting IMPDH2

Because Inhibition of IMPDH2 reduces cellular GTP level (*Ji et al., 2006*), and INZ associates with cellular IMPDH2 and reduces nucleolar NS level, consequently causing RS and p53 activation (*Figures 1–3*), we then tested if this INZ effect on p53 could be suppressed by supplementing culture media with extra GTP or guanosine. As shown in *Figure 4A–B*, addition of either GTP or guanosine to cells significantly, though partially, alleviated the INZ induction of p53 level and activity as measured by IB analysis of

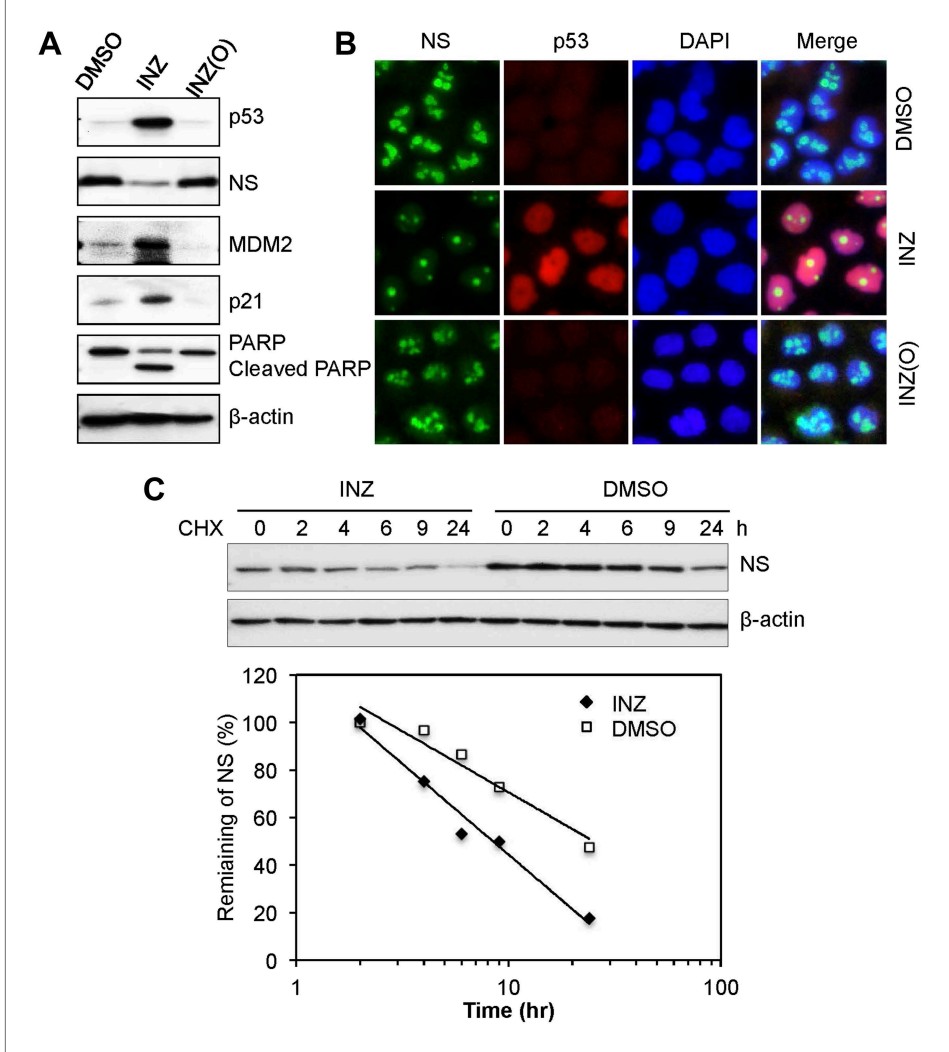

**Figure 2**. INZ, not its inactive analog INZ (O), treatment, decreases NS expression and destabilizes NS protein. H460 cells were treated with 2 μM INZ or its analogue INZ (O) for 20 hr. Cells were harvested and immunoblotted with p53, NS, MDM2, p21, cleaved PARP and β-actin (**A**), or immunostained with anti-NS (green) and anti-p53 (red) (**B**). (**C**) H460 cells were treated with 2 μM INZ for 9 hr before then 50 μg/ml of CHX was added. Cells were harvested at different time points as indicated and assayed for levels of NS.

p53, Puma and cleaved PARP. This result is well correlated with *Figure 4C*, showing that INZ markedly reduced the GTP level in H460 cells by 6.3-fold and in HCT116 cells by 3.7-fold, respectively, as measured by HPLC analysis (*Di Pierro et al., 1995*). These results indicate that INZ can reduce cellular GTP level likely by inhibiting IMPDH2 in cells. Indeed, knockdown of IMPDH2 compromised the GTP depletion by INZ treatment in both p53 wild type and p53 null HCT116 cancer cells (*Figure 4D*). Since it has been shown that NS is very sensitive to cellular GTP level and low GTP level triggers NS re-localization from the nucleolus to the nucleoplasm, consequently destabilizing it (*Tsai and McKay, 2005*; *Lo et al., 2012*), these results also suggest that it must be by decreasing cellular GTP level that INZ causes NS degradation and consequent RS, leading to p53 activation (*Figure 4E*).

Cancers are caused by alterations of multiple tumor-associated proteins or genes at the genetic and epigenetic levels (*Hanahan and Weinberg, 2011*), including the p53 pathway (*Vousden and Prives, 2009*). Thus, targeting multiple proteins of one or more signaling pathways in cancers is necessary for developing a more effective cancer therapy. Several individual SIRT1 or IMPDH2 inhibitors have been reported (*Alcain and Villalba, 2009*; *Chen et al., 2010*). However, dual targeting SIRT1 and IMPDH2 by INZ to activate p53 would offer the first paradigm for anti-cancer drug development. Our

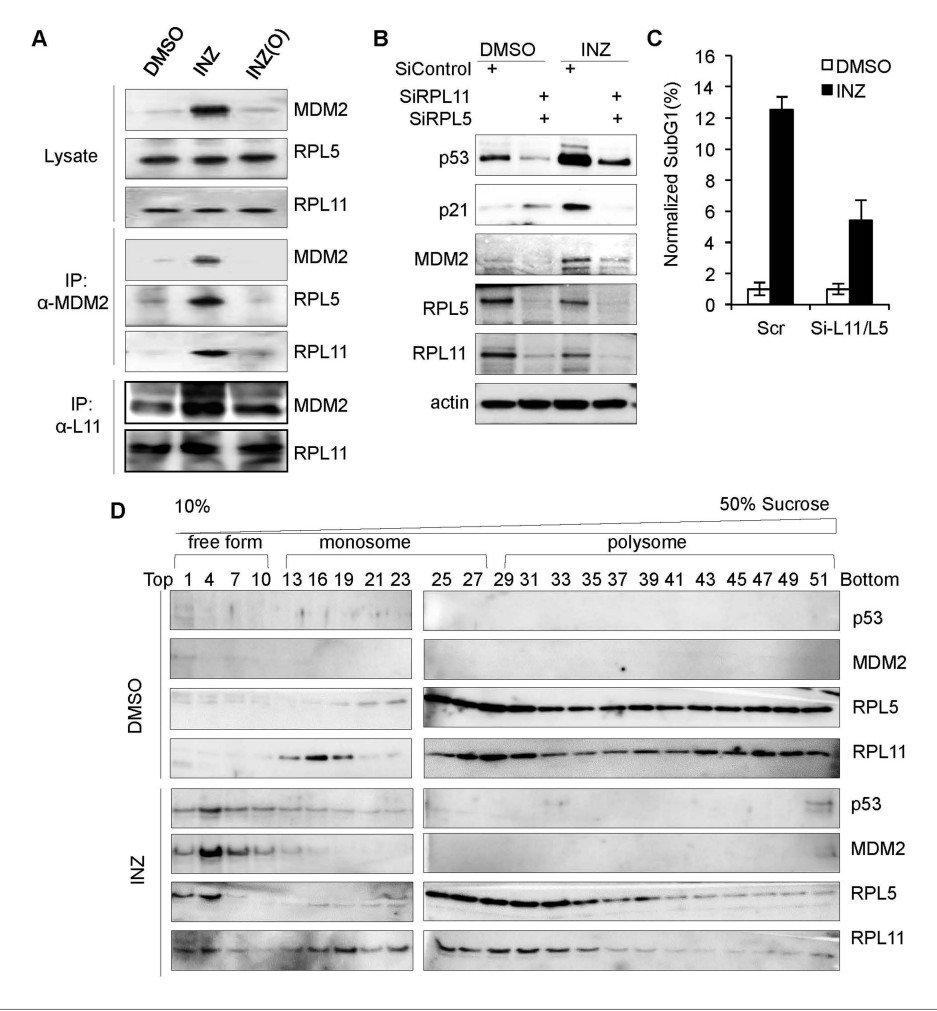

**Figure 3**. INZ treatment enhances the interaction of MDM2 with L5 and L11 by inducing ribosome-free form of RPL5 and RPL11. (**A**) H460 cells were treated with 2 μM INZ or INZ (O) for 18 hr. Cell lysates were used for IP with anti-MDM2 antibodies or anti-L11 antibodies followed by IB using anti-RPL5, RPL11 or MDM2 antibodies. (**B** and **C**) HCT116 cells were transfected with siRNAs against RPL5 and RPL11, or control, for 48–72 hr and treated with 2 μM INZ 18 hr before harvesting, followed by IB using indicated antibodies or subG1 analysis by flow cytometry. (**D**) Ribosomal profile assay. Cytoplasmic extracts containing ribosomes from H460 cells treated with or without 2 μM INZ for 18 hr were subjected to a 10–50% linear sucrose gradient sedimentation centrifugation. Fractions were collected and subjected to IB with anti-RPL11, anti-RPL5, anti-p53, or anti-MDM2 antibodies. The distribution of ribosomes is indicated.

studies (*Figures 1–4*) together with previously published findings strongly suggest that INZ effectively activates p53 and suppresses tumor growth in a p53-dependent fashion by targeting SIRT1 and IMPDH2 (*Figure 4E*, (*Zhang et al., 2012b*)). This dual targeting strategy could also explain why INZ can still partially activate p53 in IMPDH2 knockdown or GTP-supplemented cells (*Figure 1D* and *Figure 4A,B*), although the partial impairment of p53 induction could also be due to the inefficiency of completely knockdown IMPDH2 or the possible non-continuous availability of intracellular GTP throughout the experiment.

## Materials and methods

### Cell culture, reagents and antibodies

Human lung carcinoma H460 and human colon cancer HCT116 were cultured in Dulbecco's modified Eagle's medium supplemented with 10% fetal bovine serum (PBS), penicillin, and streptomycin.

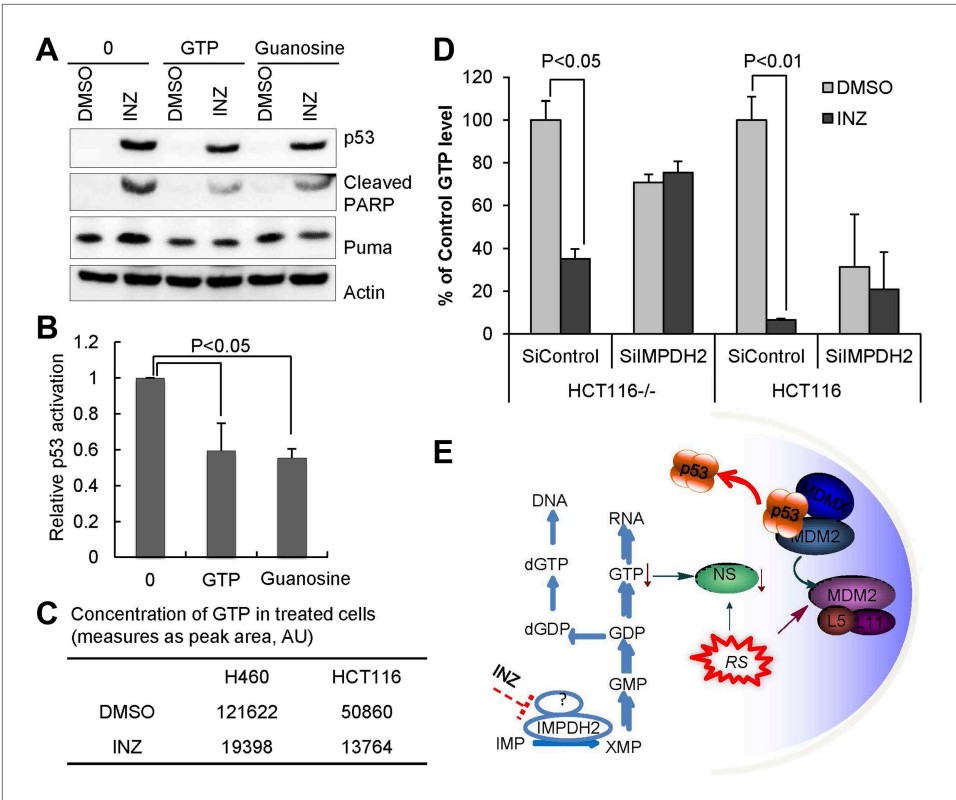

**Figure 4**. GTP or guanosine lessens INZ activation of p53 in cells. (**A** and **B**) H460 cells were pretreated with GTP or guanosine for 2 hr before the addition of 2 μM INZ. Cells were harvested and followed by IB with indicated antibodies. (**C**) Effect of INZ on cellular GTP level. The nucleotides were extracted from H460 or HCT116 cells treated with 2 μM INZ by 80% acetonitrile and SPE column. Samples were subjected to GTP analysis by HPLC. Results of quantification of HPLC spectra presented in arbitrary units (AU) were presented in this Table. (**D**) HCT116[−/−] and HCT116 cells transfected IMPDH2 siRNA (SiIMPDH2) or scrambled siRNA (SiControl) were exposed to INZ for 18 hr, and cellular GTP was extracted, measured and quantitated by LC-MS/MS. Values represent means ±SD (n = 2). (**E**) A schematic diagram of the role of IMPDH2 in INZ-induced ribosomal stress (RS) and p53 activation. IMPDH2 is a rate-limiting enzyme in the de novo guanine nucleotide biosynthesis. INZ reduced the levels of cellular GTP and NS by targeting IMPDH2 (or its complex), resulting in RS that leads to the enhancement of the RPL11/RPL5-MDM2 interaction, consequently MDM2 inactivation and p53 activation.

Inauhzin (INZ), Inauhzin inactive analogue INZ (O) (INZ9 in [*Zhang et al., 2012a*]) and Biotinylated INZs were synthesized and characterized by NMR and LC-MS as described (*Zhang et al., 2012b*). The purity of the compounds is higher than 90%. Mycophenolic acid (MPA) was purchased from Sigma-Aldrich (St.Louis, Missouri). Mouse monoclonal anti-p53 (DO-1), rabbit anti-p21 (M19), mouse anti-p21 (F5), rabbit anti-SIRT1 (H300) and goat anti-RPL11 (N17) were purchased from Santa Cruz Biotechnology, Inc. (Dallas, Texas). for immunoblotting. Cleaved PARP, PARP (9542), Puma were from Cell Signaling Technologies. Mouse anti-MDM2 (2A10), rabbit anti-RPL11 and anti-RPL5 antibodies were described previously (*Zeng et al., 1999*; *Sun et al., 2008*). Antibodies for immunostaining were rabbit poly-clonal anti-p53 (FL-393; Santa Cruz) and monoclonal nucleostemin antibodies (Chemicon, Billerica, Massachusetts).

## Biotin-avidin pull down assays

Cells were plated on 10 cm dishes and treated with compounds at about 60–70% confluence. Cells were harvested and lysed in the PBS buffer with 0.1% (wt/vol) NP40 (freshly adding protease inhibitors and 1 mM DTT). Incubate the cell lysate with 25 μL of NeutrAvidin Agarose beads (Thermo Scientific, Waltham, Massachusetts) (beads volume) in for 2 hr at 4°C with end-over-end mixing. Centrifuge at 13,000 rpm for 10 s at 4°C in a microcentrifuge. The beads were washed three times with 0.5% (wt/vol) NP-40,

0.2% (wt/vol) Tween20/Tris buffered saline and then subjected to on-beads digestion and mass spectrometry as shown below.

## On-bead Trypsin digestion

On-bead digestions were performed to release the proteins and resulting tryptic peptides from the NeutrAvidin beads. In brief, beads were resuspended in 100 μl of 50 mM ammonium bicarbonate pH 8.0 followed by the addition of 1 μg of Trypsin Gold (Promega, Madison, Wisconsin). The samples were then incubated at 37°C for 12 hr with shaking. Following digestion, samples were run through spin columns to remove any trace of the residual purification resin. The digestions were then quenched through the addition of 8 μL of formic acid.

## MudPIT analyses

Protein samples from cells that were MOCK treated with Biotin-INZ (O) or treated with DMSO or Biotin-INZ were pressure loaded onto three-phase MudPIT columns containing Aqua C18 and Luna SCX resins (Phenomenex, Torrance, California) as previously described (*Mosley et al., 2009*, *2011*). Ten-step MudPIT was performed using increasing concentrations of ammonium acetate to initiate each step followed by a 100-min gradient of 0–80% acetonitrile. All samples were analyzed on a LTQ Velos mass spectrometer (Thermo Scientific) with the dynamic exclusion set to 90 s. The spectra obtained through MudPIT analysis were searched through Proteome Discoverer 1.3 (Thermo Scientific) using SEQUEST as the peptide-spectrum matching algorithm against the Human NCBI 11-22-10 database containing 29,535 protein sequences. In addition to the human proteins, the database also contained ~140 common contaminant sequences for proteins such as keratins, BSA, and proteolytic enzymes. Using Proteome Discoverer 1.3, all peptides were required to pass a 2% false discovery threshold. The number of spectra obtained for proteins found to interact with Biotin-INZ was compared to the levels of spectra for those same proteins observed in MOCK and DMSO treatments to ensure that the candidate interacting proteins are detected at levels higher than background.

## Immunoblotting

Cells were seeded in 6-well plates. All compounds were dissolved in DMSO and diluted directly into the medium to the indicated concentrations, and 0.1% DMSO was used as a control. After incubation with the compounds for the indicated times, cells were harvested and lysed in 50 mM Tris-HCl pH 8.0, 150 mM NaCl, 5 mM EDTA, 0.5% NP-40 supplemented with 2 mM DTT and 1 mM PMSF. An equal amount of protein samples (50 μg) was subjected to SDS-PAGE and transferred to a PVDF membrane (PALL Life Science, Port Washington, New York). The membranes with transferred proteins were probed with primary antibodies followed by horseradish-peroxidase-conjugated secondary antibody (1:10,000; Pierce). The blots were then developed using an enhanced chemiluminescence detection kit (Thermo Scientific), and signals were visualized by Omega 12iC Molecular Image System (UltraLUM, Claremont, California).

## Immunofluorescence staining

H460 cells at 50–70% confluence were treated with 2 μM of Inauhzin (INZ) or Inauhzin (O) (INZ (O)) for 16 hr. Cells were fixed in 4% formaldehyde/PBS for 10 min, permeabilized and blocked with 0.3% Triton-100, 8%BSA/PBS. The primary antibodies used were monoclonal nucleostemin antibodies in 1:250 dilution and polyclonal p53 antibodies in 1: 500 dilution according to the manufactural instruction. Images were taken with a Zeiss Axiovert 200M fluorescent microscope (Germany).

## RNA interference

Control scrambled siRNA (Santa Cruz), or siRNA specific to IMPDH2 (Santa Cruz and Ambion, Grand Island, New York) were commercially purchased. These siRNAs (60 nM) were introduced into cells using METAFECTENE SI following the manufacturer's protocol (Biontex, Germany). Cells were treated with INZ for IB, cell viability assays and FACS analysis.

## Cell viability assay

To assess cell growth, the cell counting kit (Dojindo Molecular Technologies Inc., Rockville, Maryland) was used according to manufacturer's instructions. Cell suspensions were seeded at 5000 cells per well in 96-well culture plates and incubated overnight at 37°C. Compounds were added into the plates and incubated at 37°C for 72 hr. Cell growth inhibition was determined by adding WST-8 at a final

concentration of 10% to each well, and the absorbance of the samples was measured at 450 nm using a Microplate Reader (Molecular Device, SpectraMax M5e (Sunnyvale, California)).

## FACS analysis

Cells were harvested, fixed in 70% ethanol overnight and analyzed by propidium iodide (PI) staining and flow cytometry (FACS Calibur, Becton Dickinson, Washington, DC) as previously described (*Riccardi and Nicoletti, 2006*).

## Ribosomal profiling analysis

Cytosolic extractions, sucrose gradient sedimentation of polysomes, and analysis of the polysomes/mRNPs distribution of proteins were carried out as previously described (*Sun et al., 2007*; *Dai et al., 2012*; *Ingolia et al., 2012*). Briefly, cells were incubated with 100 μg/ml of cycloheximide for 15 min. Cells were homogenized in polysome lysis buffer containing 30 mM Tris–HCl (pH 7.4), 10 mM $MgCl_2$, 100 mM KCl, 0.3% NP-40, 100 μg/ml of cycloheximide, 30 units/ml RNasin inhibitor, 1 mM DTT, 1 mM phenylmethylsulfonyl fluoride, 0.25 μg/ml pepstatin A. After incubation on ice for 5 min, cell lysates were centrifuged at 1300×g at 4°C for 10 min. Supernatants were subjected to sedimentation centrifugation in a 10–50% sucrose gradient solution containing 30 mM Tris–HCl (pH 7.4), 10 mM $MgCl_2$, 100 mM KCl in a Beckman SW41 rotor at 37,000 rpm at 4°C for 2 hr. Fractions were collected and absorbance of RNA at 254 nm was recorded using BR-188 Density Gradient Fractionation System (Brandel, Gaithersburg, Maryland) to analyze the distribution of polysomes and monosomes as described (*Esposito et al., 2010*).

## Extraction and determination of cellular GTP

We adapted previously established whole cell assays using HPLC to determine cellular GTP level (*Di Pierro et al., 1995*; *Nakajima et al., 2010*). After 48 hr of growth, cells were treated with 2 μM INZ for 18 hr. Cellular nucleotides were extracted from cell monolayers by addition of ice-cold 80% acetonitrile for 1 hr. The extracts were centrifuged to pellet the cellular debris and the cleared supernatant was loaded to the SPE column (SAX column, Sigma–Aldrich). The elutes were analyzed by Agilent 1100 series liquid chromatograph system with a C18 reversed-phase column (Agilent Zorbax Extend-C18, 5 μM, 4.6 × 150 mm). A gradient elution from 0%B to 50%B in 70 min was used at a flow rate of 1 ml/min (solvent A: 0.05M $KH_2PO_4$, 0.005M tetrabutylammonium, pH5.5; B: 50% acetonitrile in 0.05M $KH_2PO_4$, 0.005M tetrabutylammonium, pH 7.0). The GTP level was also analyzed by a nano-ACQUITY UPLC/Synapt HDMS mass spectrometer (Waters, Milford, Massachusetts) using acetonitrile/water (0.05M $NH_4Ac$ buffer solution at pH = 5.5) as the mobile phase with a flow rate of 0.5 μl/min.

# Additional information

## Funding

| Funder | Grant reference number | Author |
|---|---|---|
| National Cancer Institute | CA 172468-02 | Hua Lu |
| National Cancer Institute | CA0954412-12 | Hua Lu |
| National Cancer Institute | CA127724-05 | Hua Lu |
| National Cancer Institute | CA129828-04 | Hua Lu |

The funder had no role in study design, data collection and interpretation, or the decision to submit the work for publication.

## Author contributions

QZ, Conception and design, Acquisition of data, Analysis and interpretation of data, Drafting or revising the article, Contributed unpublished essential data or reagents; XZ, Conception and design, Acquisition of data, Analysis and interpretation of data; RZW, Acquisition of data, Contributed unpublished essential data or reagents; AM, Conception and design, Acquisition of data, Analysis and interpretation of data, Contributed unpublished essential data or reagents; SXZ, ZX, Acquisition of data, Analysis and interpretation of data; HL, Conception and design, Analysis and interpretation of data, Drafting or revising the article

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
