## [Decision Letter]

Thank you for sending your work entitled “The Role of IMPDH2 in Inauhzin-induced Ribosomal Stress” for consideration at *eLife*. Your article has been favorably evaluated by James Manley (Senior editor) and 3 reviewers, one of whom is a member of our Board of Reviewing Editors.

The Reviewing editor and the other reviewers discussed their comments before we reached this decision, and the Reviewing editor has assembled the following comments to help you prepare a revised submission.

1) The authors previously showed that INZ inhibits SIRT1 and thereby activates p53. Now they provide a different mode by which this compound activates p53 and the authors do not try to consolidate these observations with their previous work showing INZ inhibits SIRT1, so we cannot assess the contribution of each activity to the end effect. When does INZ work via SIRT1 and when by inhibiting IMPDH2? Since both SIRT1 and IMPDH2 catalyze NAD+ dependent reactions is there perhaps a common mechanism by which this compound works in cells? The consequences of each function of INZ in terms of p53-independent effects are also not clear. It would be interesting to know whether INZ(O) has also lost SIRT1 inhibitory activity: could the two activities be separated using this compound?

2) What are the consequences of INZ interaction with IMPDH2? Where on IMPDH2 does INZ bind? Does it inhibit its enzymatic activity? Is that why the levels of GTP are so dramatically reduced in these cells? Are they similarly reduced in p53 null cell lines? What about normal non-transformed cells? The nature of the interaction between INZ and IMPDH2 remains unclear. Although INZ is implied in the manuscript to inhibit IMPDH2 activity, the RNAi mediated knockdown of IMPDH2 does not replicate the same effects on p53 as INZ treatment. The authors should address this uncertainty and extend the findings on the interaction between the inhibitor and IMPDH2.

More minor comments:

*Reviewer #1*:

Figure 1: IMZ dramatically induces p53 and Mdm2 but only very modestly increases p21 expression in Figure 1. Is this because the massively increased Mdm2 is degrading p21 (as the authors have previously showed), or because the impact of INZ on p53 transcriptional program is selective? Have the authors performed a kinetic analysis to show the relative peak times for p53 proteins vs p21, Mdm2 and Puma RNA and protein induction?

Figure 3: Can the authors explain why there is more p21 upon siRNA knockdown of RPL11 and RPL5 in DMSO treated cells than in INZ treated cells with these siRNAs while the opposite is seen with Mdm2 protein under the same conditions?

Figure 4 A: The data with GTP and guanosine are a bit strange in that phosphorylated nucleotides are generally not taken up by intact cells. What were the concentrations of GTP and guanosine these experiments and can authors show that adding GTP to cells increased the intracellular levels of GTP?

Only 1 siRNA is used for each knockdown in all the experiments. They need to show more siRNAs to rule out off-target effects.

Figure 3: I assume Figure 3 should be labeled Figure 3? I would recommend showing Figure E (D?) as the first panel in this experiment to set the stage for the IP shown in 3A and B.

*Reviewer #2*:

Citations should not be used in the Abstract.

Consistent denotation is needed for genes/proteins. For example TP53/p53.

Figure 1:

Legend: GRP78 is mentioned in the Figure Legend but the experiment described is actually a knockdown of IMPDH2.

1C: Treatment of H460 cells with INZ appears to decrease IMPDH2 expression whereas treatment of HCT116 cells with INZ appears to have no effect on IMPDH2 expression in Figure 1. According to the model in Figure 4, p53 is activated by a decrease in GTP mediated by decreased IMPDH2 activity; however, a decrease in IMPDH2 expression and correspondingly its activity has no observable effect on p53 activation in the absence of INZ.

1E: The Introduction states that IMPDH2 is often highly expressed in rapidly proliferating cell populations; yet, siRNA mediated knockdown of IMPDH2 causes an increase in cell proliferation.

Figure 2:

2C: The graphical representation of the half-life assay should be presented in log form so that the decrease in expression is linear and the slope can be more easily compared. Furthermore, the x-axis should be labeled in a manner that is proportional to the time between each point; for example, the distance between 2 and 4 should not be equal to that between 9 and 24.

Figure 3:

3C: Statistical significance should be added to the error bars if possible.

Figure 4:

4B: Again, if statistically significant the rescue by GTP/Guanosine treatment from vehicle should be denoted as such to strengthen the data.

4C: As a possible explanation for the lack of p53 activation in response to siIMPDH2 treatment this could be added to the GTP concentration assay to determine whether partial IMPDH2 knockdown has a comparable effect on GTP levels as INZ treatment.

*Reviewer #3*:

1) What is the effect if IMPDH2 depletion on the growth of cells without INZ treatment (Figure 1)? Are all the growth effects p53 dependent?

2) It is difficult to conclude anything about the effect of INZ on Mdm2-L11 binding in Figure 3 since the input levels of Mdm2 are so different.

---

## [Author Response]

In response to the comments and suggestions by the reviewers, we have performed a substantial amount of experiments and added 2 new panels (Figures 1 and 4) and 7 modified panel (Figures 1, 2, 3 and 4) as well as corresponding modified main text. More new results have also been obtained to address each of the reviewers’ comments. Of note, because of his help in detecting cellular GTP levels, Ruizhi Wu was added into our manuscript as a co-author.

*1) The authors previously showed that INZ inhibits SIRT1 and thereby activates p53. Now they provide a different mode by which this compound activates p53 and the authors do not try to consolidate these observations with their previous work showing INZ inhibits SIRT1, so we cannot assess the contribution of each activity to the end effect. When does INZ work via SIRT1 and when by inhibiting IMPDH2? Since both SIRT1 and IMPDH2 catalyze NAD+ dependent reactions is there perhaps a common mechanism by which this compound works in cells? The consequences of each function of INZ in terms of p53-independent effects are also not clear*. *It would be interesting to know whether INZ(O) has also lost SIRT1 inhibitory activity: could the two activities be separated using this compound?*

A) Our previous studies strongly demonstrate that INZ targets the SIRT1-p53 pathway (45). In this current study, we mainly focus on a new target of INZ, which is the ribosomal stress (RS)-p53 pathway by inhibiting IMPDH2 in cancer cells. It is possible that INZ inhibits SIRT1 and IMPDH2 in a similar fashion, since both of the enzymes utilize NAD as a cofactor. To prove that INZ can target both pathways by inhibiting SIRT1 and IMPDH2 to achieve maximum cell growth inhibition, we performed INZ dose response experiments in HCT116 cells with double knockdown of SIRT1 and IMPDH2. As shown in Figure 5, co-depletion of SIRT1 and IMPDH2 displayed an obvious combination effect on suppression of INZ-induced cytotoxicity, compared to single knockdown of either SIRT1 or IMPDH2, as their IC_50_ values increased at least 2 fold, from 3.77 to 10.57μM, and 5.20 to 10.57μM, respectively.Author response image 1.Combination effect of co-depletion of IMPDH2 and SIRT1 on INZ induced cell death. HCT116 cells, transfected with scrambled siRNA (SiControl), IMPDH2 siRNA (SiIMPDH2), SIRT1 siRNA (SiSIRT1) or co-transfected with IMPDH2 and SIRT1 siRNA, were treated with different doses of INZ and cell viability were assessed by WST cell growth assays. IC_50_ values are represented as mean ± standard deviation (n=3).

B) It had been suggested that INZ might possess p53 independent effects, for example, INZ promotes cell death in the absence of p53 at high concentrations ((19, 44), and data not shown). This could be due to other SIRT1's substrates like p73 as we showed in our previous studies. Although there is the existence of other potential protein targets (that are associated with SIRT1 and IMPDH2) for INZ, our results clearly show that this compound at lower doses specifically triggers p53-dependent apoptosis and suppression of cell proliferation in both cultured and xenograft tumors (44, 45, 46, 47) and data not shown). Indeed, knockdown of IMPDH2 by siRNA conveyed a much more significant effect on compromising the cytotoxicity of INZ in p53 wild type than in p53 null cancer cells (Figure 1), indicating that the suppression of cell growth by INZ is mainly through the inhibition of IMPDH2 and activation of the p53 pathway.

C) We used Biotin-INZ (O) for in vitro pull down assay as a negative control to identify additional potential proteins that INZ specifically targets in cancer cells. As compared to the cells treated with Biotin-INZ (O), some specific proteins were pulled down with Biotin-INZ. MS analysis revealed one of them as our previously identified SIRT1 (45) and another as IMPDH2 (Figure 2), which indicated SIRT1 and IMPDH2 are associated with INZ but not with INZ (O). This was confirmed by immunoblot analysis from both H460 and HCT116 cells (Figure 1; data not shown). We have modified the text and updated Figure 1 accordingly.

*2) What are the consequences of INZ interaction with IMPDH2? Where on IMPDH2 does INZ bind? Does it inhibit its enzymatic activity? Is that why the levels of GTP are so dramatically reduced in these cells? Are they similarly reduced in p53 null cell lines? What about normal non-transformed cells? The nature of the interaction between INZ and IMPDH2 remains unclear. Although INZ is implied in the manuscript to inhibit IMPDH2 activity, the RNAi mediated knockdown of IMPDH2 does not replicate the same effects on p53 as INZ treatment. The authors should address this uncertainty and extend the findings on the interaction between the inhibitor and IMPDH2*.

A) We performed a set of experiments to determine the effect of INZ on IMPDH2 in vitro (Figure 6). Indeed, INZ did directly bind to purified IMPDH2 in biotin-avidin pull down assays (Figure 6). Although we could pull-down IMPDH2 from cell extracts and observed the direct binding between IMPDH2 and INZ using purified proteins, INZ did not affect IMPDH2 activity using the purified enzyme (Figure 7). This discrepancy could be due to differences between the native state of IMPDH2 inside cells and the purified recombinant protein, as the former could include post-translational modifications and/or partner proteins. Another possibility is that as the chemical structure of INZ mimics a nucleoside, it could be phosphorylated by a kinase and the phosphorylated form directly inhibits the activity of IMPDH2 in vivo, as a number of nucleosideanalogues (e.g. ribavirin, mizoribine) are known to inhibit IMPDH2 after being monophosphorylated by cellular kinases (Leyssen et al 2005, Stuyver et al 2002a, Stuyver et al 2002b). Our results also suggest that INZ might not directly bind to the active site of the enzyme. Currently, we have found some interesting proteins pulled down by INZ are associated with IMPDH2 via proteomics analyses and biochemical and cell-based assays, and we have also identified the metabolites of INZ by LC-MS/MS. It is possible these metabolites of INZ and newly identified IMPDH2-binding proteins might play a role in the regulation of IMPDH2 activity by INZ. Apparently, the mechanism underlying INZ inhibition of IMPDH2 is more complex than that for INZ inhibition of SIRT1. However, studying this complex possibility would take us much longer time and generate substantial amounts of data that could be useful for another independent manuscript. Therefore, we hope that this reviewer would allow us to put these studies into our future manuscripts.Author response image 2.INZ binds to IMPDH2, but does not inhibit enzyme activity *in vitro*. (A) Recombinant IMPDH2 produced in E. coli BL21-CodonPlus (DE3)-RIPL, and purified through Ni-His columns (Lane 1) followed by TEV cleavage (Lane 2) as described in the experimental procedure. The purity of IMPDH2 enzyme (Lane 3) is confirmed before every assay by SDS-PAGE. (B) *In vitro* enzyme activity assays were conducted as described in the Supplementary Information. Mycophenolic Acid (MPA) is used as a positive control for IMPDH2 inhibition. The curve fitting and IC_50_ determination of INZ were performed using Igor Pro 4.01A. (C) Purified IMPDH2 was incubated at indicated concentrations with Biotin or Biotin-INZ that was conjugated with avidin beads. The bound IMPDH2 to Biotin-INZ was analyzed using IB with NeutrAvidin Protein-Horseradish Peroxidase Conjugated (Avidin-HRP, 1:1000; Pierce) and anti-IMPDH2 antibodies. (D) Purified IMPDH2 was incubated at indicated concentrations with Biotin-INZ or Biotin overnight at 4°C. After incubation, each mixture was subjected to Native-PAGE analysis, followed by blotting to PVDF. The blot was probed with Avidin-HRP and anti-IMPDH2 antibodies.

B) The cellular GTP level is also depleted by 60% in p53 null cells with INZ treatment (Figure 4); however, the decrease is modest compared to the over 90% depletion in the p53 wild type cells with INZ treatment. Knockdown of IMPDH2 could compromise INZ effect on the GTP levels in both p53 wild type and null cancer cells. We did not test INZ in normal cells because INZ is much less toxic to normal cells even though they contain WT p53 (44, 45)

C) The partial knockdown of IMPDH2 by siRNA and the multiple protein targets of INZ, such as SIRT1 and IMPDH2, could explain why knockdown of only IMPDH2 by siRNA does not replicate the same effects on p53 as with INZ treatment. We have repeated siRNA knockdown experiments using another specific siRNA against IMPDH2. The new immunoblots are included in the revised version (Figure 1). Although the efficiency of knockdown of IMPDH2 is better, the effect on p53 activation by knockdown of IMPDH2 is still not comparable to INZ treatment, indicating INZ targets more than one protein besides IMPDH2.

*More minor comments*:

Reviewer #1:

Figure 1*: IMZ dramatically induces p53 and Mdm2 but only very modestly increases p21 expression in*
Figure 1*. Is this because the massively increased Mdm2 is degrading p21 (as the authors have previously showed), or because the impact of INZ on p53 transcriptional program is selective? Have the authors performed a kinetic analysis to show the relative peak times for p53 proteins vs p21*, *Mdm2 and Puma RNA and protein induction?*

In our previous studies, we had shown INZ induced p53 level and transcriptional activity, in a dose and time-dependent fashion (45). The p21 expression rises significantly within 3 h, while p53 expression is noticed only after 6h from the treatment in H460 cell line. This might partially be due to p73 activation by INZ inhibition of SIRT1. Another possibility would be that p53 activity might be activated earlier than the increase of p53 steady state level in response to INZ treatment, leading to the earlier expression of p21. The time point for Figure 1 is 18h, and the MDM2 expression level is very high at that time point, and consequently, MDM2 mediates p21 degradation decreasing the level of p21. Also, please see our response to point 2 as indicated below.

Figure 3*: Can the authors explain why there is more p21 upon siRNA knockdown of RPL11 and RPL5 in DMSO treated cells than in INZ treated cells with these siRNAs while the opposite is seen with Mdm2 protein under the same conditions*?

In DMSO treated cells, the elevation of p21 level is not through transcriptional activation, because the p53 level is reduced upon RP knockdown. Therefore, the p21 expression in these cells may be regulated at the post-translational level. Many mechanisms are responsible for the regulation of p21 protein level; for example, our group and others found that MDM2 could degrade p21 independently of p53 (Jin et al 2003, Zhang et al 2004), which may partially explain the reason why p21 level in lane 2 is lower than lane 1. However, we cannot rule out other possibilities, as RPs are essential proteins, and knocking down any of them is potentially able to cause other defects. These experiments are basically designed to test whether RPs are required for INZ-induced activation of the p53 pathway. Thus we assessed the expression of two typical p53 target genes, p21 and MDM2, which indeed demonstrates that RPs are required for INZ-induced p53 activation.

Figure 4
*A: The data with GTP and guanosine are a bit strange in that phosphorylated nucleotides are generally not taken up by intact cells. What were the concentrations of GTP and guanosine these experiments and can authors show that adding GTP to cells increased the intracellular levels of GTP?* We appreciate the reviewer’s comment.

The addition of exogenous GTP or guanosine has been used to increase intracellular GTP level ([7], [20], Meshkini et al 2011). The evidence that exogenous phosphorylated nucleotides can be actively transported across the cell membrane of intact cells is indicated in the formation of phosphoproteins inside the cells (Amir-Zaltsman and Salomon 1989, Guo et al 1999, Piacentini and Niroomand 1996). For example, in labeling experiments, cells are initially placed in a specific medium designed to deplete the nucleotide pools, followed by addition of radioactive labeled 32P-ATP or GTP (or other nucleotide), leading to labeling of intracellular DNA or proteins. The mechanism is associated with the active transport by transporters of ATP or GTP located on the cell membranes (Amir-Zaltsman and Salomon 1989, Lelong et al 1992, Lelong et al 1994, Piacentini and Niroomand 1996, Wieland et al 1993).

To demonstrate that this is true to this reviewer, we also confirmed the uptake of GTP by cells using LC-MS. Cells were incubated with 100 μM GTP, and at the time point of 2hr and 4hr, the cells were thoroughly washed, and GTP was extracted, identified and quantitated by LC-MS. As shown in the figure below (Figure 7), the concentration of cellular GTP level reached the level of greater than 80 μM 4 hrs after incubation with 100 μM of GTP.Author response image 3.

*Only 1 siRNA is used for each knockdown in all the experiments. They need to show more siRNAs to rule out off-target effects*.

We appreciate the reviewer’s suggestion.

Additionally, we have purchased two different siRNA of IMPDH2 from Ambion, and repeated the knockdown experiment. The new immunoblots are included in the revised version (Figure 1 and data not shown) indicating consistent results.

Based on our experience, the siRNAs used against RPL5 and RPL11 in the current study are specific for the knockdown of these two RPs. Please also refer to our previous publications (Jin et al 2003, [19], Zhou et al 2013).

Figure 3*: I assume*
Figure 3
*should be labeled*
Figure 3*? I would recommend showing*
*Figure E*
*(D?) as the first panel in this experiment to set the stage for the IP shown in 3A and B*.

We appreciate the suggestion. We have reorganized the figure panels in Figure 3.

Reviewer #2:

*Citations should not be used in the Abstract*.

Thanks, we have corrected it.

*Consistent denotation is needed for genes/proteins. For example TP53/p53*.

Thanks, we have corrected it.

Figure 1*:*

*Legend: GRP78 is mentioned in the Figure Legend but the experiment described is actually a knockdown of IMPDH2*.

Thanks, we have corrected it.

*1C: Treatment of H460 cells with INZ appears to decrease IMPDH2 expression whereas treatment of HCT116 cells with INZ appears to have no effect on IMPDH2 expression in*
Figure 1*. According to the model in*
Figure 4*, p53 is activated by a decrease in GTP mediated by decreased IMPDH2 activity; however, a decrease in IMPDH2 expression and correspondingly its activity has no observable effect on p53 activation in the absence of INZ*.

We also observed the decrease of IMPDH2 level by INZ treatment in H460 cells but not in HCT116 cells. This is probably a cell-specific event. The depletion of IMPDH2 by siRNA is not as great as in the INZ treatment. One reason might be the low efficiency of siRNA, with the siRNA able to partially knockdown IMPDH2. Another reason might be the multiple targets of INZ besides IMPDH2. We also repeated the experiment using two additional siRNA against IMPDH2 from different vendors. Our results clearly showed that the knockdown of IMPDH2 activates p53 in HCT116 cells, although not as great as in the INZ treatment. We also determined the GTP level in IMPDH2 knockdown cells. It is obvious that the knockdown of IMPDH2 reduced GTP levels significantly in both wild type and null p53 cells. However, it was not comparable to INZ treatment, probably because depletion of IMDPH2 expression cannot activate p53 as much as INZ.

*1E: The Introduction states that IMPDH2 is often highly expressed in rapidly proliferating cell populations; yet, siRNA mediated knockdown of IMPDH2 causes an increase in cell proliferation*.

Thank you for the comment. It is possible that the scramble siRNA had side effects on cell growth somehow. We repeated the experiments using a different scramble siRNA as control. Indeed, IMPDH2 knockdown inhibited cell growth of cancer cells, as shown in the new figure panel Figure 1 (the first point indicates the absence of INZ treatment).

Figure 2

*2C: The graphical representation of the half-life assay should be presented in log form so that the decrease in expression is linear and the slope can be more easily compared. Furthermore, the x-axis should be labeled in a manner that is proportional to the time between each point; for example, the distance between 2 and 4 should not be equal to that between 9 and 24*.

Thanks for the suggestion. We have done so as suggested.

Figure 3*:*

*3C: Statistical significance should be added to the error bars if possible*.

Thanks for the suggestion. We have added P values and connecting lines for the relevant comparisons in figures throughout the paper.

Figure 4*:*

*4B: Again, if statistically significant the rescue by GTP/Guanosine treatment from vehicle should be denoted as such to strengthen the data*.

Thanks. We have done so as suggested.

*4C As a possible explanation for the lack of p53 activation in response to siIMPDH2 treatment this could be added to the GTP concentration assay to determine whether partial IMPDH2 knockdown has a comparable effect on GTP levels as INZ treatment*.

We appreciate the reviewer’s suggestion. We performed the new experiment as suggested. Knockdown of IMPDH2 significantly reduced the level of cellular GTP, however, not as strong as INZ treatment (Figure 4). This is probably one of the reasons that IMPDH2 siRNA could not reach the same effect as INZ in p53 activation.

Reviewer #3:

*1) What is the effect if IMPDH2 depletion on the growth of cells without INZ treatment (*Figure 1*)? Are all the growth effects p53 dependent*?

The depletion of IMPDH2 mediated by siRNA decreased GTP level and inhibited cell growth (Figure 1; of note, the first point indicates the absence of INZ treatment and Figure 4). The depletion of cellular GTP level and the growth inhibition by IMPDH2 depletion is much more modest in p53 null cells compared to p53 wild type cells. Please also see the above points for more details.

*2) It is difficult to conclude anything about the effect of INZ on Mdm2-L11 binding in*
Figure 3
*since the input levels of Mdm2 are so different*.

Thanks for the comment. Unfortunately, we failed to equalize the amount of MDM2 level with the addition of MG132. We then performed a reciprocal co-immunoprecipitation experiment using L11 antibody. Anti-L11 antibodies pulled down same amount of L11 in all samples, but much more MDM2 was co-immunoprecipitated with L11 antibodies in INZ treated samples (Figure 3).